# Spectrum of Thyroid Dysfunction in Patients with Chronic Kidney Disease in Benin City, Nigeria

**DOI:** 10.3390/medicines10080047

**Published:** 2023-08-09

**Authors:** John O. Obasuyi, Mathias A. Emokpae

**Affiliations:** 1Department of Medical Laboratory Science, School of Basic Medical Sciences, University of Benin, Benin City 300283, Nigeria; obasuyijohn60@gmail.com; 2Department of Medical Laboratory Service, University of Benin Teaching Hospital, Benin City 300283, Nigeria

**Keywords:** thyroid diseases, chronic kidney disease, patients, humans, Nigeria

## Abstract

There is an indication of abrupt rise in chronic kidney disease (CKD) in Nigeria and thyroid function involvement has not been sufficiently evaluated. This study determined thyroid gland function among subjects with CKD in Benin City, Nigeria. A total of 184 randomized CKD patients attending specialist clinic and 80 healthy control subjects were recruited for this study. A well-structured questionnaire was used to obtain data on socio-demography. Blood specimens were collected and used for the determination of thyroid function parameters; thyroid stimulating hormone (TSH), triiodothyronine (T3), free triiodothyronine (fT3), thyroxine (T4), free thyroxine (fT4), thyroid peroxidase antibody (TPO-Abs), thyroid globulin antibody (Tg-Abs) and Deiodinase enzyme Type 1 (D1). SPINA GD and SPINA GT were calculated using Michaelis-Menten model. The CKD was classified into stages using Modification of Drug in Renal Disease (MDRD) formula. Thyroid dysfunctions observed were clinical hyperthyroidism 1 (0.54%), non-thyroidal illness 78 (42.4%), clinical hypothyroidism 11 (6.0%), sub-clinical hyperthyroidism 3 (1.60%), and sub-clinical hypothyroidism 11 (6.0%), while euthyroid were 80 (43.5%). SPINA GD of CKD patients (33.85 ± 10.94) was not significantly different when compared with controls (24.85 ± 1.57), whereas, SPINA GT was significantly higher (*p* < 0.01) among CKD patients (3.74 ± 0.31) than controls (2.68 ± 0.11). Autoimmune thyroid disease demonstrated by positive Tg-Abs and TPO-Abs were observed among approximately 7.9% of CKD patients. Serum TPO-Abs concentration increased with CKD progression. Thyroid dysfunction is involved in the pathogenesis of CKD patients. The etiologies are multifactorial and immunological mechanisms of autoimmune thyroid disease may be a contributing factor.

## 1. Introduction

Chronic Kidney Disease (CKD) is an increasing worldwide disease affecting millions across races and continents globally. It is characterized by a progressive reduction in kidney function leading to end-stage renal disease in most subjects. The burden of renal disease in Nigeria is high, but most of the cases are under reported since there is insufficient population-based data on CKD that is required to estimate its true burden [1]. Several factors including, modifiable life style, metabolic disorders, environmental factors, amongst others have been reported to be involved in the pathways driving the pathogenesis of the disease. Chronic kidney disease is defined as sustained kidney damage evidenced by the presence of functional and or structural abnormalities such as microalbuminuria/proteinuria, haematuria, histologic abnormalities or as an unexplained decrease in glomerular filtration rate (GFR) to less than 60 mL/min/1.73 m^2^ over three months [2]. The end product of CKD is a progression to end stage renal diseases (ESRD) and development of cardiovascular disease (CKD) [3].

Thyroid hormones are very important for the physiological functions of kidneys and the maintenance of the body homeostasis. The kidneys play important roles in the breakdown and excretion of thyroid hormones. Studies have indicated thyroid dysfunction in CKD patients but the results are conflicting [4,5,6]. Chronic kidney disease has been reported to affect the hypothalamus-pituitary-thyroid axis and the peripheral metabolism of thyroid hormone, and that the frequent laboratory finding of low triiodothyronine (T3) and sub clinical hypothyroidism is the most common thyroid disorder associated with CKD [7]. It was also reported that in hyper-ureamic condition, the pituitary receptor response to thyroxine releasing hormone (TRH) is blunted leading to reduction in thyroid stimulating hormone (TSH) release. Consequently, the response of TSH to TRH is delayed because of the reduce clearance and the increase of half-life of TSH. In addition, abnormal serum constituent found in uremia can also displace T3 and thyroxine (T4) from normal protein binding site [8].

Thyroid hormones have distinguishable sequalae on cellular growth and also regulate vital physiological functions in the body. Thyroid abnormalities have been reported to increase with progression in CKD [9]. Modern laboratory testing has made the detection of minor changes in thyroid hormone status possible even though, treatment of minor alterations in thyroid hormone levels has not been recommended among CKD patients not on dialysis [9]. These minor changes of thyroid hormone levels in patients with CKD may constitute risk factor for inauspicious health consequences including CKD progression. In the same vein, CKD progression has been linked with the development of thyroid dysfunction [8].

Due to the interdependence between thyroid hormone status and kidney function as well as the lack of consistency of data on thyroid status among Nigerians with CKD, it is imperative for healthcare providers to understand the association between thyroid gland and kidney functions. This study evaluated and compared the thyroid hormone status, autoimmune thyroid disease, indices of secretory capacity of the thyroid gland (SPINA-GT) and the sum activity of peripheral deiodinases (SPINA-GD) among patients with CKD and healthy control subjects.

## 2. Materials and Methods

This is a cross-sectional study of patients with chronic kidney disease at different stages of the disease condition attending a specialist hospital (Central hospital) in Benin City, Nigeria. All patients diagnosed to have CKD following clinical presentations and Laboratory findings of hyperureamia with corresponding elevated creatinine and estimated glomerular filtration rate below 60 mL/min/1.73 m^2^ over a period of three months were enlisted in the study. A structured questionnaire was used to collect data on socio-demography and medical history.

Anthropometric data were collected by sphygmomanometer, body weight was measured with a weighing scale and recorded in kilogram while patient’s height was measured with a tape and recorded in metres. Body mass index (BMI) was calculated as the weight divided by the height square. Pubmed, Google Scholar, mpdi, academic and Ajol database repositories were utilized to search for literatures on the interaction between thyroid hormones and chronic kidney disease.

### 2.1. Inclusion Criteria

All clinically diagnosed CKD patients at different stages of the disease condition were recruited. Patients were recruited from both inpatient and outpatient departments of the medical department. Those CKD patients with serum creatinine > 5.5 mg/dL and urea level above 55 mg/dL and positive for urinary protein were included in the study based on their history, clinical signs, and symptoms. eGFR was calculated using a modification of diet in renal disease (MDRD) 4 variable formula [10].

### 2.2. Exclusion Criteria

Individuals with family history of thyroid dysfunction, currently receiving treatment for thyroid disease or taking drugs known to affect thyroid hormone indices such as glucocorticoids, salicylates, heparin, lithium, amiodarone, sulphonylurea, or phenobarbitone were excluded. Also, subjects with other disease condition (s), pregnant women and patients with known thyroid dysfunction were excluded from the study.

### 2.3. Ethical Consideration

Ethical approval (ref HM.1208/739, dated 22 July 2019) was obtained from the Ethics Committee of the Edo State Ministry of Health. Written or verbal informed consent of willingness to participate in the study was also obtained from participants.

### 2.4. Classification of Subject into Stages

Patients were grouped based on their eGFR calculated using modification of diet in renal disease (MDRD) method. The following formula was used:

eGFR (mL/min/1.73 m^2^) = 186.3 × (serum creatinine) − 1.154 × (age) − 0.203 × (0.742) if female × 1.21 if black [10,11].

### 2.5. Sample Size Determination

Sample size was determined using the sample size determination formula and a prevalence of 12.3% CKD in Nigeria [11].
$$n=\frac{Z^{n}pq}{d^{2}}$$
where: *n* = sample size, *Z*, is critical value at 95% confidence level, *p* is the prevalence, *q* = 1 − *p* and d is the precision of 5% (0.05).
3.8146 × 0.123 × 0.877/0.0025 = 166

Eleven percent attrition (18) was added to give 184 subjects. A total od 184 CKD patients and 80 healthy subjects were recruited for the study.

### 2.6. Specimen Collection and Processing

For each participant 5 mL of venous blood was collected into plain container and was allowed to clot, separated by centrifugation at 4000 rpm for 5 min. The serum obtained was stored at −80 °C for thyroid function parameters (Thyroid stimulating hormone (TSH), thyroxine (T4), free thyroxine (fT3), triiodothyronine (T3), free triiodothyronin (fT3), thyroid peroxidase antibodies (TPO-Abs), thyroglobulin antibodies (Tg-Abs) and deiodinase activity type 1 (D1). Serum TSH, T4, fT4, T3, fT3, TPO-Abs, Tg-Abs were determined using ELISA test kits supplied by Calbiotech Company, EL Cajon, CA, USA, following the manufacturer instruction while Deiodinase activities type 1 (D1), urea and creatinine were assayed by Selectra ProS manufactured by EliTech Clinical System, SAS, France, following manufacturer instructions.

The sum activity of peripheral deiodinases SPINA-GD and SPINA-GT and SPINA-GD parameters were parameters were calculated by the SPINA Thyr Software [12].

The lower and upper limits of locally generated reference ranges are
TSH = 0.35–4.50 µIU/mL;
T3 = 0.6–2.0 ng/dL;
and
T4 = 4.5–12.0 µg/dL.

The thyroid function abnormalities were defined based on the locally established reference ranges as follows:
Hyperthyroid—TSH < 0. 3 5 µIU/mL with T4 > 12.0 µg/dLHypothyroid—TSH > 4.5 µIU/mL with T4 < 4.5 µg/dLSubclinical hypothyroid—TSH > 4.5 with normal T3 and T4Subclinical hyperthyroid—TSH < 0.35 with normal T3 and T4Sick euthyroid—TSH normal with low T3.

To ensure accurate and precise results, quality control sera were assayed along with patients specimens. The control sera used were ACUSERA immunoassay premium plus level 1, 2,and 3 by RANDOX (Cat No: IA3109, IA3110, and IA3111, respectively).

### 2.7. Data Analysis

The parametric data were analyzed using Student *t*-test and analysis of variance (ANOVA), while the non-parametric data were analyzed using Chi square test. Statistical significance was set at *p* < 0.05. The statistical software INSTAT version 2.05 for window 7 (Graph pad Software Inc., La Jolla, CA, USA) was used. Graphical method was utilized to assess the normality of data before analyses.

## 3. Results

A total of 264 samples were analyzed comprising of 184 CKD patients of different stages and 80 control subjects. Of the 184 CKD patients, 74 (40.2%) were males and 110 (59.8%) were females. As for the controls, 48 (60%) were males while 32 (40%) were females. There was no significant difference (*p* > 0.05) between CKD subjects and controls in relation to gender. CKD was classified into stages according to their respective eGFR. In all, there was no CKD stage 1 in the study group, but stage 2 were 2 (1.1%), stage 3 were 22 (12.0%), stage 4 were 47 (25.5%) while stage 5 were 113 (61.4%).

The ages of the study participant ranges from 5 years to 90 years, they were however grouped in 20 years intervals. Ages 5–24 years were 12 (6.5%), 25–44 years were 33 (17.9%), 45–64 years were 77 (41.8%), 65–84 years were 53 (28.8%) and 85–104 years were 9 (4.9%). The age interval of 45–64 years has the highest numbers of CKD subjects with a percentage of 41.8%. The ages of the control group ranges from 20 years to 68 years with ages 5–24 years numbering 28 (35%), 25–44 years were 37 (46.2%), 45–64 years were 13 (16.2%), and 65–84 years were 2 (2.5%). The different in mean age between CKD patients and control subjects was significant (*p* < 0.001).

The BMI of CKD subjects were evaluated and compared with the controls. Underweight CKD subjects were 3 (1.6%), subjects with ideal body-weight were 71 (38.6%) and overweight subjects were 110 (59.8%). For the control group, 8 (10.0%) were underweight while 72 (90%) were of ideal body weight. There was no overweight individual in the control group. The comparison of the BMI between the subjects with CKD and controls was statistically significant (*p* = 0.001). The blood pressure status of subjects indicate that, 103 (56.0%) were normotensive while 81 (44.0%) were hypertensive. All control subjects were normotensive 80 (100%). Comparison of the blood pressure of CKD subject with the control group was statistically significant (*p* < 0.0001). Out of the 184 subjects with CKD, 120 (65.2%) were diabetic while 64 (34.8%) were non diabetic and comparison of the diabetic status to that of the control was significant (*p* = 0.0003).

Educationally, 44 (24.0%) of subjects with CKD had primary level of education, 62 (33.9%) had secondary education while 77 (42.1%) attained tertiary education. As for the control group, 4 (5%) had primary education, 16 (20%) attained secondary education while 60 (75%) had tertiary education. Comparison of the educational status between subjects with CKD and control subjects was significant (*p* < 0.0001). A total of 26 (14.1%) of the study participants had family history of CKD, while 158 (85.9%) has no family history of CKD, when compared to control, it was statistically significant (*p* = 0.0005) (Table 1).

Table 2 shows the comparison of the thyroid function parameters of subjects with CKD and the controls. The serum TSH, T3, T4 and fT3 were significantly lower (*p* < 0.01), and thyroglobulin antibody was significantly higher (*p* < 0.01) among subjects with CKD than controls, while fT4 was not significantly different between CKD patients and control subjects (*p* > 0.05). Although, the mean value of thyroid peroxidase antibody of subjects with CKD was higher compared with the controls, it was not statistically significant (*p* > 0.05). The activity of type 1 iodothyronine 5′-deiodinase, (D1) was significantly higher (*p* < 0.01) in CKD subjects than controls.

Table 3 shows the classification of thyroid dysfunction among subjects with CKD. It was observed that out of the 184 CKD subjects, 1 (0.54%) had clinical hyperthyroidism, 11 (6.0%) had clinical hypothyroidism, 80 (43.50%) were euthyroid, 78 (42.4%) had non-thyroid illness (NTI) or euthyroid sick syndrome (ESS), 3 (1.60%) had sub-clinical hyperthyroidism, while 11 (6.0%) presented with sub-clinical hypothyroidism.

Table 4 shows thyroid dysfunction based on stages of CKD. It was observed that among the two subjects with CKD stage 2, 1 (50%) patient had clinical hypothyroidism, while 1 (50%) was euthyroid. Among the 22 subjects with CKD stage 3, 2 (9.1%) had clinical hypothyroidism, 4 (18.2%) euthyroid, 12 (54.5%) had non-thyroidal illness status, while 4 (18.2%) had sub-clinical hypothyroidism. Among the 47 subjects with CKD stage 4, 1 (2.1%) had clinical hypothyroidism, 20 (42.5%) euthyroid, 24 (51.1%) non-thyroidal illness and 2 (4.3%) sub-clinical hypothyroidism. Of the 113 subjects with CKD stage 5, it was observed that 1 (0.9%) patient had clinical hyperthyroidism, 7 (6.2%) had clinical hypothyroidism, 42 (37.2%) had non-thyroidal illness, 3 (2.6%) had sub-clinical hyperthyroidism, 5 (4.4%) had sub-clinical hypothyroidism while 55 (48.7%) were euthyroid.

Table 5a reveals the classification of TPO-Abs in CKD subjects. It was observed that of the 184 CKD subjects, 4 (2.2%) had borderline positive TPO-Abs, 29 (15.8%) were positive while 151 (82.1%) were negative. The control group had 5 (6.5%) individual with borderline positive TPO-Abs, 10 (12.5%) positive while 65 (81.3%) were negative. The comparison of the TPO-Abs CKD subjects to the control was not statistically significant (*p* > 0.05).

Table 5b reveals the TPO-Abs classification according to CKD stages. There was no positive TPO-Abs recorded in CKD stage 2 and 3, but stage 4 had 1 (2.1%) borderline positive and 4 (8.5%) positive incidence respectively, while CKD stage 5 had 3 (2.7%) borderline positive and 25 (22.1%) positive incidence as well. The prevalence was higher in CKD stage 5.

Table 6 shows the reaction of Tg-Abs in CKD subjects. It was observed that of the 184 CKD subjects, 4 (2.1%) had borderline positive Tg-Abs, 14 (7.6%) were positive while 166 (90.2%) were negative. All control individuals were negative for Tg-Abs.

Comparison of thyroid function parameters among CKD patients at different stages of disease is shown in Table 7. It revealed that there was no significant difference in TSH among CKD stage 2 when compared with the controls, but TSH decrease significantly between CKD stage 2 and CKD stage 3, stage 4 and stage 5 respectively (*p* < 0.001). Serum T_4_ was significantly lower in all CKD stages when compared with controls *p* < 0.001. The mean value of T_4_ in CKD stage 2 (4.45 ± 0.15) was significantly lower (*p* < 0.001) compared with CKD stage 3 (6.31 ± 0.38), but there was no observable significant difference between T_4_ levels of CKD stage 4 and stage 5 respectively (*p* > 0.05). Also, fT_4_ was significantly lower in CKD stage 2 and CKD stage 3 (*p* < 0.01), compared with controls, but significantly higher as disease progresses from CKD stage 4 to CKD stage 5 (*p* < 0.01) when compared with controls.

It was observed that fT_3_ was not significantly different in CKD stage 2 when compared to controls (*p* > 0.05), but significantly lower (*p* < 0.001) in CKD stages 3, stage 4 and stage 5 (*p* < 0.0001) compared with controls. This implies that fT_3_ decreases with CKD progression.

Thyroid peroxidase antibodies (TPO-Abs) was lower in CKD stages 2, stage 3 and stage 4 when compared with controls, but the different was not statistically significant when compared with controls. Although TPO-Abs increases with disease progression, the difference was not statistically significant when compared with controls.

Even though SPINA GD of CKD subjects was higher among CKD subjects than controls, the difference was not statistically significant (*p* > 0.05). Conversely, SPINA GT of CKD patients was significantly higher than control subjects (*p* < 0.01) (Table 8). When compared with CKD stages, neither SPINA GD nor SPINA GT showed significant difference (*p* > 0.05) (Table 9).

Figure 1 shows the Heatmap illustrating the relationship between thyroid dysfunction and chronic kidney disease.

## 4. Discussion

There is an inter-play between thyroid hormone and the kidney, thyroid hormone is essential for cellular growth and differentiation as well as modulation of physiological functions in the tissues. The effect of thyroid dysfunction on renal function is attributed to the role kidneys play in the metabolism, breakdown and clearance of substances including thyroid hormones. It is envisaged that kidney dysfunction may result in altered thyroid hormone status. The need to holistically study thyroid hormone status in CKD to better understand the pathophysiology driving the disease in this aged long disorder in our clime was the motivation behind this work.

There was no gender difference between patients with CKD and controls. But more men (59.8%) than women (40.2%) were evaluated in this study. Population based studies have indicated that CKD prevalence differs by gender, affecting more women than men, but more men with CKD start renal replacement treatment than women [13,14]. The reason for the gender differences is not clear, but may be adduced to several factors;—longer life expectancy among women than men, the rapid decline in kidney function among men than women due to the protective effect of oestrogen and near absence of testosterone in women [15]. Testosterone is a possible risk factor for CKD development in men. The understanding of gender differences in the causes and mechanisms driving CKD progression may improve management strategies [15].

Most of the CKD patients were either euthyroid (43.5%) or non-thyroidal illness (42.4%). The thyroid abnormalities observed were clinical hypothyroidism (6%), clinical hyperthyroidism (0.54%), sub-clinical hyperthyroidism (1.6%) and sub-clinical hypothyroidism (6.0%). A total of 26/184 (14.1%) had thyroid hormone abnormalities. The kidney plays significant role in the excretion of iodine via the glomerular filtration. Therefore, decreased GFR in CKD patients results in a decrease iodine clearance. This may lead to an increase in plasma iodide concentration and an increase in thyroidal tissue iodide uptake. The elevated total body inorganic iodide then impairs thyroid hormone synthesis by inhibiting the pituitary thyroid axis. These alterations might describe the increased frequency of hypothyroidism in CKD patients [16]. It is important to state that majority of the patients with abnormal thyroid function were at stage 5, three at stage 4, six at stage 3 while only one patient was at stage 1. Clinical hyperthyroidism was observed in only one CKD patient at stage 5, while clinical hypothyroidism occurred in a patient at stage 2, two patients at stage 3, one patient at stage 4 and seven patients at stage 5. Subclinical hyperthyroidism was observed among three patients at stage 5, while sub-clinical hypothyroidism was observed among four patients at stage 3, two at stage 4 and five at stage 5 respectively.

The findings of greater proportion of CKD patients with non-thyroidal illness (NTI) or euthyroid sick syndrome (ESS) and euthyroid thyroid hormones in this study aligned with previous studies elsewhere [17,18,19]. These authors added that patients with advanced or progressive CKD may have changes in thyroid function test similar or consistent with the euthyroid sick syndrome; such as low to normal T4, low T3, and normal TSH levels, including low free T3 commonly observed in ESRD patients. These changes in thyroid parameters seen in CKD and ESRD patients may be due to alteration in the peripheral type 1,5′-monodeiodination of T4, as a result of increased circulating levels of some inflammatory cytokines such as tumor necrosis factor (TNF)–α and interleukin (IL)-1, metabolic acidosis, reduced levels of plasma protein, effects of drugs and presence of inhibitors of T4 binding to plasma proteins. Drugs such as furosemide and heparin (used as anticoagulant) may inhibit the binding of T4 to plasma protein and may temporally elevate free T4 concentrations which may explain the high normal fT4 reported in this study. The fT4 levels in CKD patients have been correlated with higher levels of markers of inflammation, malnutrition (lower prealbumin, IGF-1), increased endothelial dysfunction, poor cardiac function, poor survival as well as cardiovascular mortality in some studies [18,20]. Chronic Kidney Disease is a widely recognized cause of non-thyroidal illness, i.e., alterations in thyroid hormones in the absence of underlying intrinsic thyroid disorder. Low levels of the active form of the thyroid hormone, free triiodothyronine (fT3), is the hallmark of this disturbance, which is interpreted as a finalistic adaptation aimed at maintaining energy balance and minimizing protein wasting. Approximately one fourth of patients with ESRD display low fT3, thyroid dysfunction being an emerging problem also in patients with moderate to severe chronic kidney diseases. Low fT3 has recently been associated with other diseases apart from CKD, including diabetes mellitus and cardiovascular disease (CVD) [18,21].

In this study, non-thyroidal illness prevalence was also high across the stages of the disease progression, though there was none recorded in stage 2 which probably may be due to few patients investigated. The stage to stage stratification of thyroid dysfunction, CKD stage 3 had 54.5% incidence of non-thyroidal illness; stage 4 recorded 51.1% while stage 5 had 37.2% compared to hypothyroidism and hyperthyroidism. This is not consistent with that of Pan et al. [18], who reported that the most common thyroid dysfunction in CKD is non-thyroidal illness which increased with increasing stage of CKD disease rising to 69.1% in CKD stage 5.

The observed frequency of CKD patients with concurrent autoimmune thyroid disease in this study 14/184 (7.6%) for Tg-Abs and 29/184 (15.8%) for TPO-Abs are partially close to 82/1032 (7.9%) for Tg-Abs and TPO-Abs reported in a retrospective study in China [22]. The CKD patients with concurrent autoimmune thyroid disease had higher levels of membranous nephropathy and focal segmental glomerulosclerosis, reduced mesangial proliferative glomerulonephritis compared with CKD patients without autoimmune thyroid disease [22]. Autoimmune thyroid disease has been associated with hyperthyroidism and hypothyroidism. Most systemic immune diseases are linked to damage of several organs [23]. The fact that most of the CKD patients in this study had no concurrent autoimmune thyroid disease showed that there are other etiological causes of CKD. Other authors reported 20% CKD patients with hypothyroidism were TPO-Abs positive while 80% were negative and 74% of the control subjects were positive for TPO-Abs [24]. This is also partially consistent with the present study.

It was observed that the levels of TPO-Abs increased with increasing progression of CKD. Most of the positive Tg-Abs and TPO-Abs cases were observed among CKD stages 3–5. This aligned with previous study [25]. The authors reported significantly higher risk of cardiovascular disease among patients with autoimmune thyroid disease than those without autoimmune thyroid disease. It is therefore important to pay more attention to modifiable factors so as to slow down kidney disease progression and CVD among patients [25]. The findings from this study is not consistent with that of Basu and Mohapatis [8], who reported very low incidence of autoimmune thyroid disease among CKD patients. The serum TPO-Abs is important in clinical setting for the early detection of thyroid hormone disturbance. This may reduce the long-term morbidity of CKD and other associated health challenges.

Serum T_4_ was significantly lower in all CKD stages when compared with controls (p< 0.001). This observation agreed with previous study [10]. The authors reported that low total T_4_ levels with increased free T_4_ levels are seen in patients undergoing haemodialysis as heparin inhibits T_4_ binding to proteins, thereby increasing a free T_4_ fraction in these patients. Conversely, the finding did not agree with that of Aryee et al. [9], who reported that FT4 level was significantly higher in CKD patients than in the control group.

It was observed that the activity of 5′-deiodinase (D1) among CKD patients was significantly higher than controls but the activity did not varied significantly between CKD stages. The kidney is the organ endowed with the most abundant deiodinase activity which is type 1 (D1). The increase in D1 observed in this study across the stages may be due to its released into the peripheral circulation following renal damage and probably a compensatory mechanism to produce more T3 to compensate for the already low level [18].

Although SPINA GD among CKD subjects were higher than that of the control individuals, there was however no significant difference when compared with controls (*p* > 0.05). Dietrich et al. [12] reported that since the dissociation constant of type 1 deiodinase (D1) is beyond physiological serum levels of fT4, SPINA GD is nearly linear in the euthyroid range, so that it has similarities to the T3/T4 ratio. Its non-linear properties are advantageous especially in cases of high fT4 concentrations.

Conversely, SPINA GT among CKD patients were significantly higher than that of the control subjects (*p* < 0.01). SPINA GT also known as thyroid capacity or thyroid output gives an estimate of the maximum secretion rate of the thyroid gland under stimulated conditions. SPINA GT is able to clearly distinguish between functional thyroid disorders of primary origin and euthyroidism. It is unaffected by hypothalamic-pituitary dysfunction unlike TSH. The observation in this study suggest the existence of a thyroid-mediated TSH-T3 shunt, which might represent a compensatory mechanism, mitigating the effects of decreasing thyroid output in onset hypothyroidism [26].

When compared with CKD progression, SPINA GD was observed to be significantly higher as disease condition worsen (*p* < 0.01). This may suggest increase deiodination activity with CKD progression which may likely be a compensatory role to produce more T3 due to hypothyroidism to maintain the euthyroid state. There was no significant increase in SPINA GT in relation to CKD progression.

The Limitations of this study are that the subjects were recruited from a single centre, which may introduce crowd bias. However, the study evaluated patients with chronic kidney disease irrespective of the causes of CKD thereby presenting credible results. It was not possible to exclude all confounding factors and causal association may not be possible.

The clinical utility of the findings of the present work is that; although studies that have evaluated the use of thyroid hormone supplementation and its resultant effect in CKD patients are limited, it was been suggested that undertreated and untreated CKD patients with thyroid dysfunction might have a higher mortality risk than those with euthyroid function.

## 5. Conclusions

Thyroid gland dysfunction was observed in 26/184 (14.1%) among CKD patients while 78/184 (42.4%) had euthyroid sick syndrome and 80/184 (43.5%) were euthyroid. The pattern of thyroid dysfunction observed were 11/184 (6.0%) clinical hypothyroidism, 11/184 (6%) sub-clinical hypothyroidism, 3/184 (1.6%) sub-clinical hyperthyroidism and 1/184 (0.50%) clinical hyperthyroidism. Autoimmune thyroid disease demonstrated by positive Tg-Abs and TPO-Abs were observed among approximately 7.9% of CKD patients. Serum TPO-Abs concentration increased with CKD progression. Thyroid dysfunction is involved in the pathogenesis of CKD patients. The etiologies are multifactorial and immunological mechanisms of autoimmune thyroid disease may be a contributing factor.

## Figures and Tables

**Figure 1 medicines-10-00047-f001:**
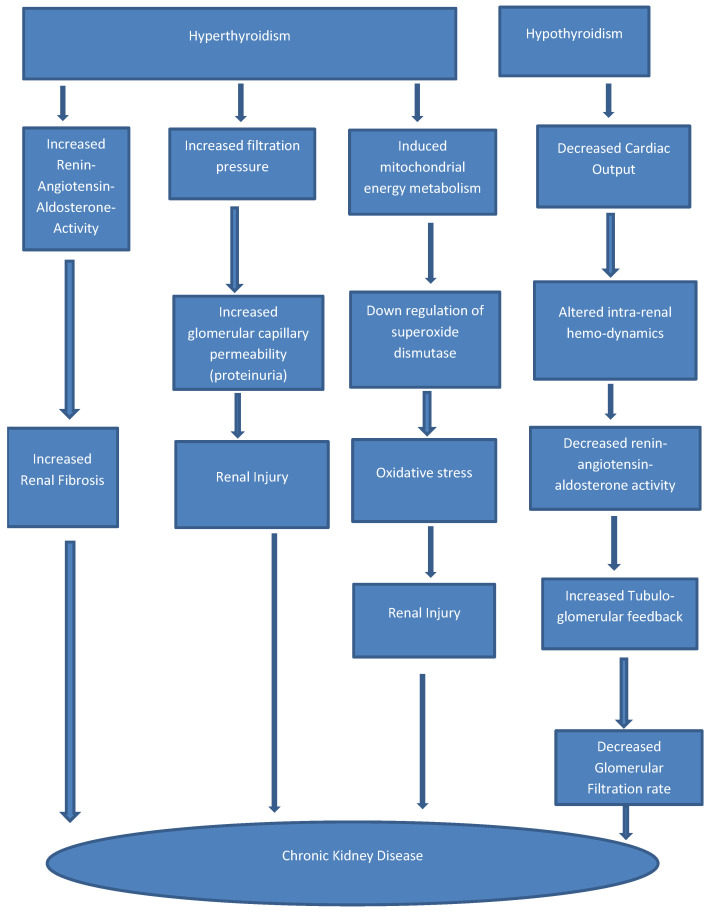
Heatmap illustrating the relationship between thyroid dysfunction and chronic kidney disease.

**Table 1 medicines-10-00047-t001:** Socio demography and Anthropometric status of the study Participants.

Variables	Category of Subjects CKD Patients N (%)	Controls N (%)	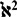	*p*-Value
SEX	Female	74 (40.2)	32 (40.0)	0.0011	0.5426
Male	110 (59.8)	48 (60.0)
Age of Subjects	5–24 years	12 (6.5)	28 (35.0)	79.853	<0.0001
25–44 years	33 (17.9)	37 (46.2)
45–64 years	77 (41.8)	13 (16.2)
65–84 years	53 (28.8)	2 (2.5)
85–104 years	9 (4.9)	0 (0) *
BMI	Underweight	3 (1.6)	8.0 (10.0)	13.14	0.0014
Normal Weight	71 (38.6)	72 (90.0)
Over Weight	110 (59.8)	0 (0.0)
Hypertension Status	Normotension	103 (56.0)	80 (100)	78.757	<0.0001
Hypertension	81 (44.0)	0 (0.0)
Educational Status of Subjects	Primary	44 (24.0)	4 (5%)	26.26	<0.0001
Secondary	62 (33.9)	16 (25%)
Tertiary	77 (42.1)	60 (75%)
Diabetics Status	Normal	64 (34.8)	80 (100)	11.551	0.0003
Diabetic	120 (65.2)	0 (0.0)
Family History	NO	158 (85.9	80 (100)	10.998	0.0005
YES	26 (14.1)	0 (0.0)
CKD Stages	Stage 2	2 (1.1)	0 (0.0)	NA	
Stage 3	22 (12.0)	0 (0.0)	NA	
Stage 4	47 (25.5)	0 (0.0)		
Stage 5	113 (61.4)	0 (0.0)		

CKD = Chronic Kidney Disease; BMI, Body Mass Index, NA—not applicable. * Not represented, percentage in parenthesis.

**Table 2 medicines-10-00047-t002:** Comparison of Thyroid function parameters between subjects with CKD and Controls.

Parameters	CKD Patients (*n* = 184)	Controls (*n* = 80)	*p*-Value	Level of Significant
TSH (µIU/mL)	1.58 ± 0.09	3.05 ± 0.08	0.000	***p* < 0.01**
T4 (µg/dL)	5.72 ± 0.14	10.2± 0.31	0.000	***p* < 0.01**
fT4 (ng/dL)	1.99 ± 0.85	0.90 ± 0.05	0.207	*p* > 0.05
T3 (ng/dL)	0.72 ± 0.03	1.34 ± 0.09	0.000	***p* < 0.01**
fT3 (pg/mL)	1.74 ± 0.07	2.51 ± 0.13	0.000	***p* < 0.01**
TPOAb (IU/mL)	31.38 ± 2.88	28.91 ± 2.99	0.553	*p* > 0.05
Tg-Ab (IU/mL)	34.59 ± 4.92	18.09 ± 2.14	0.002	***p* < 0.01**
D1 (ng/mL)	19.14 ± 1.70	2.80 ± 0.79	0.000	***p* < 0.01**

FSH = thyroid stimulating hormone; T4 = thyroxine, fT4 = free thyroxine; T3 = triiodothyronine; fT3 = free triiodothyronine; TPO-Ab = thyroid peroxidase antibody; Tg-Ab= thyroid globulin antibody and Di = iodothyronine 5′-deiodinase Type 1. Values in mean ± SEM.

**Table 3 medicines-10-00047-t003:** Classification of thyroid dysfunction among subjects with CKD.

Classification of Thyroid Dysfunction	CKD PatientsN (%)	ControlsN (%)	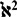	*p*-Value
Clinical Hyperthyroidism	1 (0.54)	0 (0.00)	NA	NA
Clinical Hypothyroidism	11 (6.00)	0 (0.00)		
Euthyroid	80 (43.50)	80 (100.00)		
Non-Thyroidal Illness	78 (42.40)	0 (0.00)		
Sub-clinical Hyperthyroidism	3 (1.60)	0 (0.00)		
Sub-clinical Hypothyroidism	11 (6.0)	0 (0.00)		

CKD = chronic kidney disease; percentage in parenthesis.

**Table 4 medicines-10-00047-t004:** Thyroid dysfunction based on stages of CKD.

Thyroid Dysfunction	CKD Stages			
	Stage 2	Stage 3	Stage 4	Stage 5
	N (%)	N (%)	N (%)	N (%)
Clinical hyperthyroidism	0 (0)	0 (0)	0 (0)	1 (0.9)
Clinical hypothyroidism	1 (50)	2 (9.1)	1 (2.1)	7 (6.2)
Euthyroid	1 (50)	4 (18.2)	20 (42.5)	55 (48.7)
Non-thyroidal illness	0 (0)	12 (54.5)	24 (51.1)	42 (37.2)
Sub-clinical hyperthyroidism	0 (0)	0 (0)	0 (0)	3 (2.6)
Sub-clinical hypothyroidism	0 (0)	4 (18.2)	2 (4.3)	5 (4.4)

Percentage in parenthesis.

**Table 5 medicines-10-00047-t005:** (a) Comparison of TPO-Abs positive cases between CKD subjects and controls. (b) Positive TPO-Abs based on CKD stages.

**(a)**
**TPO-Abs Reaction**	**CKD Patients**	**Controls**	** 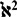 **	***p*-Value**
**N (%)**	**N (%)**
Borderline positive	4 (2.2)	5 (6.3)	3.123	0.2098
Negative	151 (82.1)	65 (81.3)		
Positive	29 (15.8)	10 (12.5)		
TOTAL	184 (100)	80 (100)		
**(b)**
**TPO-Abs Reaction**	**CKD Stage 2**	**CKD Stage 3**	**CKD Stage 4**	**CKD Stage 5**	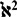	***p*-Value**
**N (%)**	**N (%)**	**N (%)**	**N (%)**
Borderline positive	0 (0) *	0 (0) *	1 (2.1)	3 (2.7)	3.123	0.2098
Negative	2 (100) *	22 (100) *	42 (89.4)	85 (75.2)		
Positive	0 (0) *	0 (0) *	4 (8.5)	25 (22.1)		
TOTAL	2 (100)	22 (100)	47 (100)	113 (10)		

Percentage in parenthesis; * Not included in statistical analysis.

**Table 6 medicines-10-00047-t006:** Tg-Abs positive Reaction among Patients with CKD controls.

Tg-Abs Reaction	Subjects N (%)	ControlsN (%)	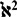	*p*-Value
Borderline positive	4 (2.1)	0 (0)	NA	NA
Negative	166 (90.2)	80 (100)		
Positive	14 (7.6)	0 (0)		
TOTAL	184 (100)	80 (100)		

N = number of cases, percentage in parenthesis; Tg-Abs = Thyroglobulin antibodies. NA: Not applicable.

**Table 7 medicines-10-00047-t007:** Comparison Thyroid Function Parameters among CKD Patients at different CKD Stages.

Parameter	CKD Stages				Controls *n* (80)	*p*-Value
	Stage 2 *n* (2)	Stage 3 *n* (22)	Stage 4 *n* (47)	Stage 5 *n* (113)		
TSH (µIU/mL)	a	b	c	d	e	<0.0001
	2.90 ± 2.40	1.30 ± 0.24	1.46 ± 0.19	1.67 ± 0.11	3.05 ± 0.09	
T4 (µg/dL)	4.45 ± 0.15	6.31 ± 0.38	5.69 ± 0.30	5.63 ± 0.17	10.24 ± 0.31	<0.0001
fT4 (ng/mL)	0.26 ± 0.25	0.74 ± 0.10	1.10 ± 0.09	2.63 ± 1.39	0.90 ± 0.05	<0.0001
T3 (ng/dL)	0.80 ± 0.40	0.52 ± 0.06	0.75 ± 0.09	0.74 ± 0.03	1.34 ± 0.09	0.001
fT3 (pg/mL)	2.40 ± 0.20	1.67 ± 0.20	1.67 ± 0.13	1.77 ± 0.09	2.51 ± 0.13	<0.0001
TPOAbs (IU/mL)	10.00 ± 5.00	17.68 ± 2.22	24.40 ± 4.15	37.32 ± 4.25	28.91 ± 2.99	0.0569
Tg-Abs (IU/mL)	35.00 ± 34.0	36.32 ± 9.65	26.49 ± 5.10	37.61 ± 7.48	18.09 ± 2.14	0.3043
D1 (ng/mL)	13.15 ± 2.65	23.75 ± 5.76	18.80 ± 2.95	18.49 ± 2.23	2.80 ± 0.80	<0.0001

Values in mean ± SEM; TSH = Thyroid Stimulating Hormone, T4 = Thyroxine, fT4 = Free Thyroxine, T3 = Triiodothyronine, fT3 = Free Triiodothyronine, TP0-Abs = Thyroid Peroxidase Antibody, Tg-Abs = Thyroid Globulin Antibody, D1-Deiodenase activity type 1.

**Table 8 medicines-10-00047-t008:** Comparison of SPINA GD and SPINA GT between Patients with CKD and Controls.

Parameters	CKD Subjects *n* (184)	Controls *n* (80)	*p*-Value	Level of Significant
SPINA GD (nmol/s) (Ref. range 20–40 nmol/s)	33.85 ± 10.94	24.85 ± 1.57	0.4164	*p* > 0.05
SPINA GT (pmol/s) (Ref. range 1.4–8.7 pmol/s)	3.74 ± 0.31	2.68 ± 0.11	**0.0015**	***p* < 0.01**

Figures are mean ± SEM.

**Table 9 medicines-10-00047-t009:** SPINA GD and SPINA GT among CKD Patients at Different CKD Stages.

Parameters	Stages 2 *n* = 2	Stages 3 *n* = 22	Stages 4 *n* = 47	Stages 5 *n* = 113	*p*-Value	Level of Significant
SPINA GD (nmol/s)	10.80 ± 2.10	17.64 ± 4.72	24.80 ± 11.30	24.93 ± 6.03	0.9544	>0.05
SPINA GT (pmol/s)	2.31 ± 0.455	3.6 ± 1.66	3.96 ± 1.13	3.37 ± 0.89	0.2159	>0.05

Figures are mean ± SEM.

## Data Availability

Data from PhD project conducted at the Department of Medical Laboratory Science, University of Benin, Benin City, Nigeria.

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
