# Peer review of "Spectrum of Thyroid Dysfunction in Patients with Chronic Kidney Disease in Benin City, Nigeria"

_medicines, 2023, doi:10.3390/medicines10080047_

Round 1

Reviewer 1 Report

"The article titled 'Spectrum of Thyroid Dysfunction in Patients with Chronic Kidney Disease in Benin City, Nigeria' is highly relevant to the journal. The manuscript demonstrates excellent organization and well-written content. Furthermore, the experimental results exhibit high quality.

During the review process, I have several suggestions and recommendations:

1. Please ensure that at least five keywords are provided to accurately represent the study's content.

2. Could you kindly provide a heatmap illustrating the relationship between thyroid dysfunction and the various stages of chronic kidney disease (CKD)?

3. I would appreciate more information about the clinical and historical features considered in your study.

4. When discussing the observed thyroid gland dysfunction in CKD patients, it would be valuable to specify the specific stage(s) of CKD where this was observed (e.g., stage 1, stage 2, etc.).

5. Are there any limitations associated with this study that should be acknowledged and addressed?

6. It would greatly enhance the study if the authors mention the specific database repository utilized for this research."

Author Response

Reviewer 1:

Comments and Suggestions for Authors

"The article titled 'Spectrum of Thyroid Dysfunction in Patients with Chronic Kidney Disease in Benin City, Nigeria' is highly relevant to the journal. The manuscript demonstrates excellent organization and well-written content. Furthermore, the experimental results exhibit high quality.

During the review process, I have several suggestions and recommendations:

  1. Please ensure that at least five keywords are provided to accurately represent the study's content. (Response thyroid diseases; chronic kidney disease; patients; humans; Nigeria)
  2. Could you kindly provide a heatmap illustrating the relationship between thyroid dysfunction and the various stages of chronic kidney disease (CKD)? (Response: Provided below)
  3. I would appreciate more information about the clinical and historical features considered in your study.(Response clinical and historic features of patients are presented in table 1)
  4. When discussing the observed thyroid gland dysfunction in CKD patients, it would be valuable to specify the specific stage(s) of CKD where this was observed (e.g., stage 1, stage 2, etc.(Response: Clinical hyperthyroidism was observed in only one CKD patient at stage 5, while clinical hypothyroidism occurred in a patient at stage 2, two patients at stage 3, one patient at stage 4 and seven patients at stage 5. Subclinical hyperthyroidism was observed among three patients at stage 5, while sub-clinical hypothyroidism was observed among four patients at stage 3, two at stage 4 and five at stage 5)
  5. Are there any limitations associated with this study that should be acknowledged and addressed? (Response: Limitation of this study are that the subjects were recruited from a single centre, which may introduce crowd bias. However, the study evaluated patients with chronic kidney disease irrespective of the causes of CKD thereby presenting credible results. It was not possible to exclude all confounding factors and causal association may not be possible.
  6. It would greatly enhance the study if the authors mention the specific database repository utilized for this research."(Response: For literature review, Pubmed, Google Scholar, mpdi, academic and Ajol database repositories were utilized to search to search for literatures on the interaction between thyroid hormones and chronic kidney disease)

Reviewer 2 Report

Authors studied thyroid hormones in relation with chronic kidney disease. The paper sound scientific and ethical enough. However, few polishing is required. In methodology, authors stated that 'The parametric data were analyzed using Student t-test and analysis of variance (ANOVA), while the non-parametric data were analyzed using Chi square test.' Both ANOVA and student t test are being used in comparison of variables that fit into normal distribution. However, authors should not assume that the study parameters to fit into homogeneous distribution. Rather, they should perform a normality analysis and must state it appropriately in methodology.

Another issue is that both thyroid disorders (Mal J Med Health Sci. 2021;17(1):101-104) and chronic kidney disease (Frontiers in Medicine, 2021, 8: 642296.) are associated with anemia. Could authors provide more data about the hemoglobin level of the participants?

Finally, I recommend authors to comment on clinical utility of the findings of the present work.

Minor spelling errors and typos should be revised. (i.e. 'body weight was measure' to 'body weight was measured').

Author Response

Reviewer 2

Comments and Suggestions for Authors

Authors studied thyroid hormones in relation with chronic kidney disease. The paper sound scientific and ethical enough. However, few polishing is required. In methodology, authors stated that 'The parametric data were analyzed using Student t-test and analysis of variance (ANOVA), while the non-parametric data were analyzed using Chi square test.' Both ANOVA and student t test are being used in comparison of variables that fit into normal distribution. However, authors should not assume that the study parameters to fit into homogeneous distribution. Rather, they should perform a normality analysis and must state it appropriately in methodology.(Response graphical method was utilized to assess the normality of data before analyses)

Another issue is that both thyroid disorders (Mal J Med Health Sci. 2021;17(1):101-104) and chronic kidney disease (Frontiers in Medicine, 2021, 8: 642296.) are associated with anemia. Could authors provide more data about the hemoglobin level of the participants? (Response: Yes, this is correct but this was not the focus at the beginning of the study)

Finally, I recommend authors to comment on clinical utility of the findings of the present work. (Response: Even though studies that have evaluated the use of thyroid hormone supplementation and its resultant effect in CKD patients are limited, it was suggested that undertreated and untreated CKD patients with thyroid dysfunction might have a higher mortality risk than those with euthyroid function)

Comments on the Quality of English Language

Minor spelling errors and typos should be revised. (i.e. 'body weight was measure' to 'body weight was measured').

Reviewer 3 Report

It is very important things in clinically settings.

I have a few comment.

Low T3 and T4 concentrations have been associated

with increased mortality in patients with CKD,

especially from cardiovascular causes from

previously report.

How about meaning of the elevation of serum TPO-

Abs concentration in clinically setting? Need to check

up routinly?

Thyroid dysfunction is occuring women rather than

men. Should you need to do multivariable analysis

for eliminating adjustment factors. 

Author Response

Reviewer 3:

Comments and Suggestions for Authors

It is very important things in clinically settings.

I have a few comment.

Low T3 and T4 concentrations have been associated with increased mortality in patients with CKD,

especially from cardiovascular causes from previously report.

How about meaning of the elevation of serum TPO-Abs concentration in clinically setting? Need to check

up routinely?(Response: The serum TPO-Abs is important in clinical setting for the early detection of thyroid hormone disruption. This may reduce the long-term morbidity of CKD and other associated health challenges).

 Thyroid dysfunction is occuring women rather than

men. Should you need to do multivariable analysis for eliminating adjustment factors.(More men than women were evaluated in this study) 

Thank you for choosing Medicines to publish your paper.